# A Qualitative Study on the Design and Implementation of the National Action Plan on Antimicrobial Resistance in the Philippines

**DOI:** 10.3390/antibiotics11060820

**Published:** 2022-06-17

**Authors:** Maria Margarita M. Lota, Alvin Qijia Chua, Karen Azupardo, Carlo Lumangaya, Katherine Ann V. Reyes, Sharon Yvette Angelina M. Villanueva, Helena Legido-Quigley, Evalyn A. Roxas

**Affiliations:** 1College of Public Health, University of the Philippines Manila, 625 Pedro Gil St., Ermita, Manila 1000, Philippines; knazupardo@up.edu.ph (K.A.); c_lumangaya@icloud.com (C.L.); smvillanueva4@up.edu.ph (S.Y.A.M.V.); earoxas1@up.edu.ph (E.A.R.); 2Saw Swee Hock School of Public Health, National University of Singapore, Singapore 117549, Singapore; alvin.chua@nus.edu.sg (A.Q.C.); ephhlq@nus.edu.sg (H.L.-Q.); 3Alliance for Improving Health Outcomes, Inc., 62 West Avenue, Quezon City 1104, Philippines; kannvillegas@gmail.com

**Keywords:** antimicrobial resistance, implementation plan, policy development, policy analysis, One Health, Philippines

## Abstract

Antimicrobial resistance (AMR) is a global public health threat that warrants urgent attention. Countries developed their national action plans (NAPs) following the launch of the Global Action Plan on AMR in 2015. The development and implementation of NAPs are often complicated due to the multifaceted nature of AMR, and studies analyzing these aspects are lacking. We analyzed the development and implementation of the Philippine NAP on AMR with guidance from an AMR governance framework. We conducted in-depth interviews with 37 participants across the One Health spectrum. The interviews were transcribed verbatim and were analyzed thematically, adopting an interpretative approach. The enabling factors for NAP implementation include (1) a high level of governmental support and involvement of relevant stakeholders, (2) the development of policies to support improved responses in infection prevention and control and antimicrobial stewardship, and (3) better engagement and advocacy by professional associations and civil society groups. The challenges include (1) a lack of resources and regulatory capacity, (2) insufficient impetus for AMR research and surveillance, and (3) limited One Health engagement. Although there has been considerable progress for human health, strengthening the involvement and representation of the animal health and environment sectors in the AMR scene must be undertaken. Developing well-defined roles within policies will be paramount to the strong implementation of AMR strategies.

## 1. Introduction

In recent years, antimicrobial resistance (AMR) has emerged as one of the major global public health threats. Human activities have been shown to contribute to the evolution and spread of AMR in the environment [1]. The misuse and abuse of antimicrobials in human and veterinary medicine are major contributing factors. These activities include the overuse of antimicrobials with variable quality and efficacy, the improper storage and disposal of antimicrobials, and the overuse of preservatives (for both food and antibacterial agents) in household products [2,3]. This problem is compounded by the following factors that persist in various countries: a lack of access to appropriate antimicrobial therapy, weak regulation in antimicrobial use (AMU) for humans and in animal production and aquaculture, self-medication of antimicrobials, weak AMR and AMU surveillance systems, slow updates on AMU treatment guidelines, unsystematic continuing medical education on AMU for prescribers, and unregulated disposal of industrial waste and waste in dumps [4,5,6,7]. In Asian countries, AMR has been a growing concern over the last few decades. This is a result of various factors, including unregulated drug distribution [8], unethical marketing practices [9], and poor animal husbandry [10], which are shaped by wider issues, such as conflict and underfunding of the public sector.

Further, it is alarming that AMR is increasing when the development of novel antibiotics is slowing down. This poses a risky situation in which some infections could no longer be treated, leading to loss of life, especially in resource-limited countries [11]. Consequently, the World Health Organization (WHO) launched the Global Action Plan on AMR in 2015, focusing global attention on the urgent need to design a multi-sectoral national action plan (NAP) on AMR [12]. At present, several countries have already established their respective plans. However, there is limited opportunity to investigate and inform the engagement process between international policy bodies and national stakeholders. A review of the evidence for national AMR policies highlighted that the evidence is limited and context-specific. In particular, there is a critical dearth of studies from Asia, which, therefore, constrains the understanding of the political and economic context in designing the effective development of appropriate interventions [4,13]. To date, no qualitative study has been conducted to analyze the development and implementation of AMR policies in the Philippines. Therefore, our study aims to discuss the policy process of developing and implementing the NAP on AMR via in-depth interviews (IDIs) to identify policy lessons and the best practices to guide advocacy regarding AMR policy.

### 1.1. AMR Situation in the Philippines

The Philippines’ experience in addressing AMR started over three decades ago, as early as 1988, with the establishment of the Antimicrobial Resistance Surveillance Program (ARSP). This is a laboratory-based surveillance program that facilitates the consolidation of data from sentinel sites and prepares the national and sub-national AMR surveillance reports. Through the ARSP, the country provides annual surveillance reports on the existing patterns of AMR in the country to various agencies, particularly to the WHO’s Global Antimicrobial Surveillance System. There have been two NAPs on AMR (2015–2017 and 2019–2023) issued in the past decade.

The 2015 NAP to Combat AMR was developed to provide the country strategies and aimed to control the emergence of AMR [14]. It ensured the implementation of an integrated, comprehensive, and sustainable national program to combat AMR geared towards safeguarding human and animal health while preventing interference in the agricultural, food, trade, communication, and environmental sectors. Apart from engaging multiple sectors, its other strategies include: (1) strengthening surveillance and laboratory capacity, (2) ensuring uninterrupted access to safe and quality-assured antimicrobials, (3) regulating and promoting the optimal use of antimicrobials, (4) implementing appropriate measures to reduce infection across all settings, (5) promoting innovation and research on AMR, and (6) improving awareness and understanding of AMR through effective communication and education [15]. This provided structure and direction to healthcare facilities in adopting a proactive multidisciplinary approach to combat AMR. National health policies and programs were eventually implemented within a devolved setting. Local government units (LGUs) were expected to carry out health promotion and prevention activities. Subsequently, the 2019 NAP built on key strategies identified in the 2015 NAP. Similar to other Asian countries, multisectoral engagement and collaboration in the manner of a One Health approach across the human health, animal health, and environmental sectors have also been emphasized in the recent NAP in view of the need to involve more sectors and agencies in the program implementation [16].

### 1.2. Conceptual Framework

Chua et al. previously published an AMR governance framework after an analysis of ten AMR NAPs from the member states of the Association of Southeast Asian Nations (ASEAN) [17]. This framework (Figure 1) was used to guide the research team during the research process. The framework highlights governance domains on policy design, implementation tools, monitoring and evaluation, and sustainability, with One Health engagement situated at the center as it influences all the other domains.

## 2. Materials and Methods

### 2.1. Study Design and Population

This is a qualitative study involving in-depth interviews (IDIs). Data collection, including recruitment and interviews, was conducted from October 2020 to May 2021. Potential participants were purposively sampled to capture a wide range of responses across the human health, animal health, and environmental sectors. These stakeholders were from the government sector (national and subnational policymakers), medical and veterinary health bodies, regulatory authorities, pharmaceutical industry, animal feed industry, environmental groups, and civil society groups. Participants were approached for an IDI via email and phone calls explaining the purpose of the research. The invitation was deemed as rejected if no response was received after two reminder emails. The research team reached out to 108 potential participants, and a total of 37 individuals agreed to participate (Table 1). The majority of the 37 respondents represent the human health sector (56%), while the rest are divided among the animal health (16%) and environmental (13%) sectors. One respondent represented both the human and animal health sectors. The other 71 potential participants did not respond to the invitation and were deemed to have declined participation. This may be attributed to the focus of the different sectors on mitigating the impact of the COVID-19 pandemic along with other issues at the time of data collection (e.g., outbreak of African swine fever). In some instances, the researchers were instead referred to the relevant policies on AMR for their sector.

### 2.2. Data Collection

The interviews were conducted by five researchers (M.M.M.L., E.A.R., S.Y.A.M.V., K.A.V.R., and K.A.) who had prior qualitative research training. A total of 33 interviews were completed, with 37 respondents participating in the research. Two of the sessions had multiple respondents because they belonged to the same program or organization and opted to participate together. There was no prior engagement nor established relationship between the researchers and the interviewees. The interviews were conducted virtually over Zoom in view of the COVID-19 pandemic, with only the researcher and interviewee present. Each interview was one-off and lasted an hour on average. These were each audio-recorded and transcribed verbatim. A semi-structured question guide was used to explore our research aim based on participants’ areas of expertise and areas of interest. The question guide (Appendix A) was developed with questions focusing on (1) knowledge and perceptions about AMR; (2) questions about their role in AMR, exploring political, cultural, economic, and institutions interests/values; and (3) their experiences of the development and implementation of the NAPs, identifying challenges and lessons learnt in the Filipino context. Participants were not remunerated for the interview.

### 2.3. Analysis

All audio recordings were transcribed verbatim in English. Interviews that were not in English were translated from Filipino to English before transcription. Identifying data were removed from all research documents to maintain confidentiality. Each transcript was coded with an interview number and the relevant One Health sector so that extracts from the same individual can be linked. QSR NVivo software (Release 1.5.1) was used for data management and sharing among team members. We adopted an interpretative approach that focuses on participants’ perceptions and interpretations of the discussion topic. We used Chua’s AMR governance framework to deductively guide our coding process while allowing codes that did not fit into the framework to develop as emerging themes inductively. Coding was conducted in an iterative manner, where we performed line-by-line analysis, constantly reassessing documents coded initially and comparing emerging codes. The first round of coding was performed by at least two research associates (A.Q.C. and K.A.) independently, followed by a second round completed by M.M.M.L., E.A.R., S.Y.A.M.V., and K.A.V.R. All researchers agreed that we were able to provide a thorough analysis of all the themes identified. Thematic saturation was established when we identified ‘rich information’ in each of the themes and we discussed and agreed that no new themes were emerging from the data. Findings were reported according to the COREQ checklist (Appendix A).

### 2.4. Ethical Considerations

Ethical approval was obtained from the University of the Philippines Manila Research Ethics Board (UPMREB Code 2020-485-01) and the National University of Singapore, Saw Swee Hock School of Public Health Department Ethics Review Committee (SSHSPH-038). Our protocol adheres to the Data Privacy Act of 2012 (Republic Act 10173, 2012) and the National Ethical Guidelines for Health and Health-Related Research 2017 (PHREB, 2017) of the Philippines. All respondents were provided an information sheet and were asked to review and sign a consent form, or, if they are unable to provide a signed copy, to indicate their consent through email or before the interview process started. Consent was also obtained for audio-recording the session. Respondents were able to ask questions and express their concerns at any point during the interview. Each respondent kept a copy of both the information sheet and consent form for participation in the study. Identifying information was removed to ensure anonymity and confidentiality. In addition, confidentiality was ensured by giving each respondent the option of not being quoted, even anonymously, in the study and subsequent publications. All research-related files were password-protected and access was restricted to the research team.

## 3. Results

The findings are presented under five main themes in accordance with the five domains in Chua’s AMR governance framework: policy design, implementation tools, sustainability, monitoring and evaluation, and One Health engagement.

### 3.1. Policy Design

The themes under the policy design domain included participation, accountability and coordination, strategic vision, transparency, and equity.

#### 3.1.1. Participation

The action against AMR was steered by an Inter-agency Committee on AMR (ICAMR) led by the Department of Health (DOH) and the Department of Agriculture (DA), which formulate policies on AMR and ensure and regulate access to quality and affordable antimicrobials [18]. The member agencies include the Department of the Interior and Local Government, which ensured the implementation of these AMR policies across LGUs; the Department of Science and Technology, which ensured the prioritization of AMR in the national health and research agenda; and the Department of Trade and Industry, which focused on enhancing surveillance and diagnostic capacity. It should be noted that activities within the environmental sector have yet to be integrated into the structure of the ICAMR at the time of writing.

The NAP was developed in consultation with stakeholders from the ICAMR and their attached agencies, as well as professional associations from the animal and human health sectors, representatives from academia, and, to a lesser extent, the pharmaceutical industry. These organizations were consulted regarding the plan and were invited to subsequent meetings. The stakeholders would also hold sectoral or similar meetings to discuss AMR. They would also coordinate with regulatory agencies, such as the Philippine Food and Drug Administration (FDA), regarding the implementation of rules and regulations. However, most respondents felt that more could be done to improve the engagement of the various stakeholders.


*“What needs to be improved is the active engagement of the other ICAMR agencies, and also to expand more, to engage more sectors to take part in AMR initiatives so that the Action Plan becomes more holistic.”*
(Human Health—04, International NGO)

In particular, most respondents from the environment sector said they were aware of some of the plans and programs but were not directly involved. However, the current work priorities of environmental science specialists did not necessarily connect with AMR policymaking. Several respondents perceived this as unintentional since the priorities of the environmental sciences were different, and they proposed solutions to improve their involvement:


*“What I see in my opinion, there is a little disconnect between the environmental sciences and the policymaking bodies about AMR. So, it is not intentional, but … since the priority of the environmental sciences is different… AMR seems to be masked by other priorities, and so if the policymaking bodies could also tap us in proposing projects that could also deal with the AMR, then I think our involvement will start there.”*
(Environmental Health—01, Academia)

#### 3.1.2. Accountability and Coordination

Given the scale of coordination work and resources required for the NAP, the proponents endeavored to have official government support in the form of an Administrative Order.


*“Our first target to produce, even before an action plan, was to have a policy, so that the policy signed by the Office of the President will make it easier for us to mobilize funds.”*
(Human Health—08, Government)

The ICAMR was the primary body for coordination and implementation. The secretariat was with the Pharmaceutical Division of the DOH. Some respondents also associated NAP accountability with the DOH due to its visibility compared to the DA. If certain agencies were unable to meet the targets, their representatives were requested to discuss their plans or proposals for resolving the gaps. In a devolved setting, the local implementation of AMR-related programs was bridged by sub-national offices of the respective government agencies to ensure that these efforts contribute to national targets. For example, the DA formed regional councils and technical working groups on AMR.


*“Definitely on their role, we meet the TWG (technical working group), say DA. After that, we present the output to stakeholders through consultation meetings. There are other topics as well wherein even in the initial meetings the stakeholders were already there. We present the idea to them and (listen to) what they can say about their contribution.”*
(Animal Health—03, International NGO)

From a One Health perspective, one pain point was the regulation of veterinary drugs and products that were mainly used by stakeholders governed by the DA. The regulation of these products, however, was being transferred to the FDA, which is an attached agency of the DOH. Coordinating the lines of regulation over products on the one hand and activities on the other would be a challenge, according to several respondents who were aware of the developments in this area. The Joint Administrative Order was readopted in 2020, which sets a transition period of six months for the transfer of the following functions to the FDA: the regulation of veterinary drugs and products, veterinary biological products, and establishments.


*“In some parts, from the BAI (Bureau of Animal Industry), they have, I would say, lower-level regulations. Because previously there was an MOU (Memorandum of Understanding)… regarding this regulation, and it expired. When it expired, there was some gap on how to implement.”*
(Animal Health—02, International NGO)

Although relevant department orders have been issued, most respondents emphasized the need for intersectoral coordination to address inappropriate antibiotic use, along with other control measures.


*“…there are a lot of very good department orders that they have issued, both DOH and DA, but not sufficient emphasis was put up in terms of the coordination efforts, and addressing the misuse, the overuse, underuse of all these antibiotics plus all the other controls that must be addressed…”*
(Human Health—12, National Organization)

The fragmented nature of the Philippine health system, stemming from the devolution of health services to LGUs, presented challenges to both policymakers and implementers [19]. In terms of policy development, most of the stakeholders noted that the roles of the LGUs, civil societies, and other sectors must also be clearly articulated.


*“In this country, the main challenge is (that) it is fragmented… Another thing, the standard is rich LGUs, rich provinces, always have better quality than poor ones… now with antibiotic restriction, at the minimum, you need a doctor, so no doctor in an LGU means no antibiotic.”*
(Human Health—09, National Organization)


*“I also think that greater emphasis should be given in coordination with the LGUs because both DA and DOH were devolved when the Local Government Code of 1991 was passed… I think the next thing that should happen is they should mobilize the LGUs, the civil societies.”*
(Human Health—12, International NGO)

#### 3.1.3. Strategic Vision

A strategic vision through clearly defined goals and objectives is necessary for the development of effective plans. The NAP envisions “a nation protected against the threats of antimicrobial resistance” using “a multifaceted and holistic strategy to consolidate the fragmented efforts and systems in the country by enabling different stakeholders such as health professionals, policymakers, and other government systems to use it as a platform for national campaigns on prudent antibiotic use” [14,16].

In the latest NAP (2019–2023), the ICAMR aimed to maximize the One Health approach and further clarify and align the mandates, roles, and responsibilities of the member agencies. The body also emphasized the need to harmonize policies, protocols, and guidance, and to “continue capacity building, education, awareness and advocacy” [16]. In particular, priorities for the year are discussed and agreed upon by the DOH and DA to secure resources for the implementation.


*“I would say both sectors, the DOH and the Department of Agriculture. They agreed on how to set up the priorities for the year, (and) on what they will be working (on). And based on those agreed priorities, and based of course on the allocation of resources, I know both of them. They also get a special fund from the government to implement the National Action Plan, apart from the external donor support. They agree on these certain priority activities and agree to implement at the same time.”*
(Animal Health—02, International NGO)

In animal health, there were four key areas, namely: (1) AMR and antimicrobial use (AMU) awareness, (2) surveillance, (3) governance, and (4) good practices. The Food and Agriculture Organization provides support to the DA in the development of AMR plans. The components of the plan in animal health were AMR surveillance in healthy food animals by the National Meat Inspection Service; pathogens in diseased animals, including livestock and poultry, by the Bureau of Animal Industry; and aquaculture and fisheries by the Bureau of Fisheries and Aquatic Resources.

It was reported that the NAP development also planned for monitoring targets, which were mostly centered around the reduction in, or maintenance of, the prevalence of resistant organisms. Other targets included a reduction in the antibiotic use in humans and animals and identification of the baseline AMR and antibiotic use in animals (Table 2).

#### 3.1.4. Transparency

Among the ICAMR member agencies, there were discussions on the progress and milestones related to the strategies identified in the NAP. The Antimicrobial Resistance Surveillance Program (ARSP) has been consistent in publishing annual reports on AMR surveillance, while the DOH Pharmaceutical Division also provides progress on AMR through the agency’s annual report, all of which are publicly available.


*“There are published surveillance reports yearly… we come up with the annual report by May of the succeeding year, this is because… data would be in by January of the succeeding year, so we also need time for analysis, verify, and publish.”*
(Human Health—17, Government)

Some stakeholders expected greater transparency in the AMR policy process, with reports as one of the main outputs they would like to have access to. There were a number of reporting requirements for the national AMR Program, but not all reports are made publicly available. A few respondents, ranging from patient representatives to local implementers, mentioned that there was no feedback on the information requested from them. More importantly, most reported that they would like to have clarity on success indicators and how they, as stakeholders, can contribute. One participant from the environmental sector shared similar sentiments.


*“…hopefully on the policymakers’ end, we get updates or we get the latest information on what is the status already, and then what do we have to watch out for?…What are the indicators of success? Again, it has not been disclosed to us or shared with us.”*
(Human Health—11, National Organization)

Similarly, it was reported that the lack of auditing impacts the management and quality of information.


*“There is no audit… The reporting side is very poor. In fact, if it were not made into a rule that it could be punishable, the reportable diseases, there would be no one to report. There is also no one to audit. So if there’s no report and there’s no audit, how do you get the information? This is one of the big gaps.”*
(Animal Health—07 National Organization)

#### 3.1.5. Equity

Most stakeholders held the view that one of the AMR interventions should be to help to promote access to healthcare as this means access to services and products, including safe, quality, and effective antimicrobials. In some barangays or rural health units, it was reported that there are efforts to implement the Botika ng Barangay, literally “village drugstore”, which was a strategy that expanded access to essential drugs based on current standards and regulations.


*“One thing that’s very clear with us is that the public health interventions that we want to put in terms of AMR, we don’t want them ultimately to become anti-access.”*
(Human Health—15, International NGO)

In the human health sector, the most mentioned focus was on understanding the perspectives of patients, consumer groups, and civil society because these things affected access and the quality of health services. The Joint Administrative Order issued by the DA and the FDA was among the initiatives helping to improve access to healthcare products and services. The FDA was tasked with issuing marketing authorizations and licenses to operate and sell antimicrobials for human and veterinary use.

### 3.2. Implementation Tools

The themes under the implementation tools domain included surveillance, infection prevention and control (IPC), optimizing AMU, AMR research, education and public awareness, and international collaboration.

#### 3.2.1. Surveillance

AMR surveillance was led by the Research Institute for Tropical Medicine (RITM) and was limited to selected tertiary hospitals and laboratories that conduct sentinel surveillance. Based on a respondent from the government, lower-level hospitals could only collect and transport specimens of suspected cases to these sentinel sites.

A respondent reported that the surveillance activities for animals were irregular and usually only conducted upon request. A few respondents also noted that there were reporting delays.


*“You can have more data in humans but there is still a vacuum or limited data for animals. If we see humans, animals and then (the) environment. Because we don’t see that.”*
(Animal Health—04, Academia)

The Bureau of Animal Industry has been mobilized to develop their existing surveillance for AMR in animals, led by an AMR team within the bureau. Three components have been identified by the participants for AMR surveillance in animals, which entail collaboration and coordination among multiple agencies. The National Meat Inspection Service, the Bureau of Animal Industry, and the Bureau of Fisheries and Aquatic Resources were involved in the sampling of (1) food animals, (2) diseased animals, and (3) fish and seafood, respectively. The residues in pork and beef were reportedly being monitored in partnership with academic institutions.

According to a few respondents, there were no mechanisms in place for the environmental monitoring of AMR. Only a few laboratories had the capacity for these surveillance activities, with the data primarily generated by research groups. Additionally, in the animal and environmental health sectors, several standards and guidelines were still being reviewed or not firmly in place even if surveillance is being conducted. For example, as the next quote illustrates:


*“Environmental health per se and surveillance, it’s not yet established… there’s still so much to do. There’s still no standards, so how would you say that the water is safe? Let’s say you were able to identify residues, what are the safety levels? We still do not have that.”*
(Human Health—04, International NGO)

AMU surveillance has been reported to be lacking as well. A few participants mentioned that, in hospital settings, there was no system in place to support the monitoring, evaluation, and regulation of antibiotics that were subsidized by PhilHealth [20]. The National Meat Inspection Service conducted the surveillance of veterinary drug residues and quarterly monitoring of AMR in slaughterhouses in the National Capital Region, an initiative launched in 2018. The Bureau of Fisheries and Aquatic Resources oversaw the national residue monitoring and control program for aquaculture [20]. However, according to an independent implementation review in 2019, there was “no formal document on the monitoring of AMR and AMU food trends in food-producing animals.” The report also noted that “quality monitoring of the prescription and control of veterinary drugs is an area for concern.”

#### 3.2.2. Infection Prevention and Control

In 2009, the DOH published the National Standards in Infection Control for Healthcare Facilities to strengthen the IPC programs nationwide and prevent healthcare-associated infections. In 2016, priority strategies were defined in the national policy on IPC in healthcare facilities, including the institutionalization of an antimicrobial stewardship (AMS) program [21]. The Infection Control Committee serves as the multidisciplinary body for cooperation and information sharing and ensures the implementation of infection control strategies by formulating and updating infection control policies, guidelines, and procedures.

The Infection Control Committee is composed of “management, physicians, and other healthcare workers from clinical microbiology, pharmacy, sterilizing service, housekeeping, and training services” [21]. IPC training and related indicators were part of the hospital licensing requirements [16]. In animal health, the harmonization of standards and practices was being pursued at the ASEAN level. Several respondents also said that the recent spate of animal disease outbreaks, such as ASF, has led to the heightened promotion and practice of IPC.

Vaccination of patients and healthcare workers has been cited by some respondents as an essential strategy against infections. Vaccines were procured by the National Health Department at the centralized level and distributed to the sub-national offices. As part of the National Immunization Program, it was mentioned that primary vaccines were given to vulnerable populations. In addition, healthcare workers might be provided with free vaccines from their respective institutions. In the following quote, an interviewee from an International Non-governmental Organization highlights the importance of “up-to-date vaccination” in reducing the incidence of infection:


*“Reducing the incidence of infection is the third key strategic objective of the Global Action Plan, which is also adopted in the National Action Plan. When we talk about reducing the incidence of infection, that’s where vaccination comes in, keeping vaccinations up-to-date, and infection prevention and control measures.”*
(Human Health—04, International NGO)

Meanwhile, several participants highlighted that good housing and engineering in farms were promoted to prevent infections among farm animals, such as poultry or pigs. However, a respondent involved in animal production said that vaccination in food animals usually depends on client requirements:


*“… my concern was the advent of several diseases which we cannot identify, many of them among our farms and the level of mortality was high. So we asked them, the hatchery practitioners, what vaccines do you use? When do you use the vaccines in your hatchery? I will tell you, their answers varied. Like, it depends on what their client wants.”*
(Animal Health—06, National Organization)

Several respondents held the view that the government should strengthen the monitoring of IPC measures. Although there is a national plan for infection control and education, most respondents believe that hospitals must be made more accountable as there were a scarcity of data on healthcare-associated infections.

#### 3.2.3. Optimizing Antimicrobial Use

Optimizing AMU involves the implementation of AMS strategies and regulations to ensure reductions in inappropriate AMU.

With the support of the World Health Organization (WHO), the Philippines became one of the pilot sites for the implementation of the AMS program in hospitals. According to the ICAMR, this program has been considered for adoption in other countries [16].

The AMS program was developed to “strengthen the knowledge, attitude and practices of involved stakeholders on rational prescribing, dispensing and use of antimicrobials; and, to improve patient outcomes by decreasing infections caused by resistant organisms” [21]. Specifically, it aimed to promote rational and optimal antimicrobial therapy, and facilitate behavioral and institutional changes in the optimal use of antimicrobials by relevant stakeholders. It also aimed to address sustainability through “multidisciplinary leadership and commitment, clinical governance and accountability in antimicrobial management and control” and to create a supportive environment in which healthcare professionals have the tools and systems to implement antimicrobial management. With an AMS monitoring tool for hospitals, the monitoring and evaluation of the AMS program were required to be documented and “shall always be available for public health purposes” [21].

Most respondents had a favorable view of AMS, especially at the primary care level as it helps to ensure access to the appropriate drugs. If AMS was strictly followed, then there should be no problems with misuse, according to several respondents.

As a participant emphasized, the success of AMS implementation also relied on the ability to train all health professionals and all stakeholders in both public and government health institutions.


*“The implementation’s goal really is to be able to train all health professionals and all stakeholders in both public and government health institutions, so that is the main goal, that all of us will be together in fulfilling and achieving AMS.”*
(Human Health—14, Academia)

Although tertiary hospitals had personnel trained regarding AMS, it was mentioned that there was a lack of infectious disease doctors and clinical pharmacists in most hospitals, which raises the question of how these facilities approve restrictions and monitor AMS.

On the regulation front, several respondents discussed how the “no prescription, no dispensing” policy (Republic Act 10918) enabled the FDA to issue a notice of violation to pharmacies that sell antimicrobials without a prescription. In the next quote, a participant believes that the DOH “no prescription, no dispensing” policy improved the dispensing of antibiotics:


*“In terms of preventing the misuse or overuse of antimicrobials, I would say the no prescription, no dispensing policy of the DOH somehow made a dent and improved the dispensing of antibiotics.”*
(Human Health—04, International NGO)

However, some also mentioned that patients were still able to buy antibiotics from “sari-sari” stores (small community stores) or obtain them from LGUs. For over-the-counter medicines, these stores were also required to apply for authorization from the FDA. In addition, it was reported that the “no prescription, no antibiotic” policy was not effectively implemented in the animal and agriculture sectors. The over-the-counter sale of antibiotics remains, according to our interviewees, one of the biggest issues in AMR for the animal health sector.

In terms of completing antibiotic therapy, some participants mentioned there were cases in which patients discontinued their antibiotic regimen once they felt better in order to save money. Some respondents added that the Philippine Health Insurance Corporation (PhilHealth) should reimburse drugs, including selected and restricted antimicrobials, under the National Formulary, but they were unsure about the stage of implementation.

#### 3.2.4. AMR Research

In terms of research focused on AMR, it was reported that the funders are mostly from advanced economies, particularly in environmental health. Programs such as veterinary medicine also collaborate with institutions abroad, but these opportunities have reportedly been infrequent. One participant mentioned that the situation was such that basic data on AMR and relevant indicators were not readily available or monitored, making it difficult to generate complementary research data using techniques such as genomic sequencing.


*“I would like to have the data to see. You are going to base your data on WASH, but do we have the WASH data? How many wastewater treatment plants can we really survey? Do municipalities actually use wastewater treatment plants? Are wastewater systems integrated? So, even the basic stuff, we still don’t have data for them, that’s why I am thinking that this will take a long time because if we don’t have the data for basic sanitation how can we get a higher-level genomic sequencing for these data sets.”*
(Environment—02, International NGO)

#### 3.2.5. Education and Public Awareness

Most stakeholders reported that education and public awareness on AMR still needed substantial improvement in several areas, such as the proper use of antibiotics. However, campaigns such as the annual World Antimicrobial Awareness Week and its national counterpart, the Philippine Antimicrobial Awareness Week, had reportedly been instrumental in disseminating information on AMR, broadening awareness and support for the national and global AMR campaign. An example provided was the campaign on the “no prescription, no dispensing policy”.

The respondents cited that other platforms being used for raising awareness and information dissemination include social media, webinars, conferences, and learning modules. Some awareness initiatives were conducted through social media platforms (e.g., TikTok) and student competitions, such as poster-making contests. Social media was viewed favorably, largely because it appealed to the younger generation.


*“because*
*its (social media’s) reach is wide, it gets a lot of traction also for the younger people”*
(Environment—02, International NGO)

Some participants mentioned that there was growing support for AMS in the form of professional development programs for community and hospital pharmacists. Online training sessions are regularly conducted for physicians, including infectious disease specialists, residents, and general practitioners, on the proper prescription of antimicrobials. It was mentioned that some training opportunities, including those on AMR and AMS, were supported by drug companies.


*“… drug companies still have quite a lot of influence… There are a lot of so-called interventions in terms of educational activities, but many of them are sponsored by drug companies”*
(Human Health—16, National Organization)

Some interviewees mentioned that there were efforts to broaden the geographic scope of training, with some provided by sub-national hospitals in the hope that they would cascade and equip other facilities and stakeholders in the region.

Several respondents said that, in general, it was more challenging for the animal sector because of the complexity and scope of issues involved. Professionals in the veterinary sector said they recognized their role as educators. The animal producers supported the AMR advocacy because they understood that the issue affected their bottom line.


*“You will hear from them, the backyard raisers, that regulation is difficult in the animal health sector. I think in terms of awareness-raising, they are the ones that are still not being reached.”*
(Human Health—04, International NGO)

A respondent pointed out the impact of limited information and engagement with related businesses in the animal sector, which was a major source of environmental contamination. The next quote highlights the need for education versus just having a punitive approach:


*“We really need proper education. Let us make it easier for them [animal producers]. It cannot be all regulation then pay the penalties. You have not even started your business [but] you are already at a loss. And everything is costly.”*
(Animal Health—06, National Organization)

Some respondents reported that AMR was still not a core subject in environmental science programs. A respondent explained that the closest topics were water quality modeling and hydrology, which also tackle bacteria in water. Overall, as the next quote illustrates, there is a perception that there is still not an in-depth understanding of environmental AMR:


*“In-depth understanding on environmental AMR is not there yet.”*
(Environment—01, Academia)

#### 3.2.6. International Collaboration

From an AMR governance perspective, the Philippines was described by a respondent as a showcase country at the regional and global levels. The country’s efforts on the ICAMR and on the environmental health agenda were viewed as instrumental in generating funding support. In fact, the ARSP at RITM was one of the earliest programs on surveillance for AMR. To support this area, a respondent said there was a need to:


*“Understand our current surveillance systems in terms of resistance and antibiotic use, and how we can then build on the current system”*
(Human Health—10, Government)

Several respondents identified international donors as the biggest funding source of the national AMR program, especially during its formative years. The country situation analysis and AMS training package were among the first initiatives with global partners. As a development partner, a few respondents reported that the WHO provides some funding support for the AMR program, primarily to establish guidelines, such as the AMS guidelines for hospitals and guidelines for primary healthcare.


*“We have a lot of technical support coming from the tripartite. So it is just a matter of agencies having a serious commitment to implement.”*
(Human Health—10, Government)

In providing the needed financial resources (e.g., grants), it was mentioned that the development partners made it possible for the program to gain momentum despite resource constraints. In addition, the country’s AMR awareness campaign was reportedly based on the annual World Antimicrobial Awareness Week held every second week of November, which carries the WHO theme. While several respondents noted that there was some government funding for this activity through the ICAMR, it was reported that the investment from each sector remains minimal.

### 3.3. Monitoring and Evaluation

Monitoring and evaluation involve reviewing the effectiveness of implementation plans, reporting them to relevant stakeholders, and developing feedback mechanisms for future improvements.

#### 3.3.1. Effectiveness and Reporting

At the point of writing, surveillance was largely on the monitoring targets defined in the NAP (Table 2). For human health, the RITM was the main institution in charge of AMR surveillance, which was limited to selected sentinel sites. The coverage of AMU monitoring was limited to hospitals and pharmacies. Although there was a review conducted on the status of the implementation of the plans from the 2015 NAP, there was no structured system in place for regular monitoring of the other implementation plans [20]. For example, some respondents highlighted that stakeholders do not have a mechanism to assess the effectiveness of education and awareness initiatives and how these activities have improved AMR strategies.


*“So I would say that it’s a continuous engagement to implement by the government. In terms of accountability, if they could not comply or achieve what is in the NAP, what would happen? So it’s being reported, that’s it.”*
(Animal Health—02, International NGO)

Based on the RITM 2020 Annual Report, there were fewer isolates reported by bacteriology laboratories compared to 2019. The data showed that vancomycin and linezolid resistance continue to be reported for enterococci, and the rates of methicillin resistance in *Staphylococcus aureus* were on the decline. Multidrug-resistant *Escherichia coli*, *Klebsiella pneumoniae*, *Acinetobacter baumanii*, and *Pseudomonas aeruginosa* persisted, while penicillin resistance in *Streptococcus pneumoniae* was increasing.

#### 3.3.2. Feedback Mechanism

Prior to the COVID-19 pandemic, the participants mentioned regular monitoring meetings at the ICAMR level involving the government, the industry, the producers, the academia, the research groups, and other stakeholders. This provided a platform for evaluation and feedback, which informed areas for improvement in policy and program implementation. The next quote illustrates how meetings are organized:


*“Normally what we do is, there’s an agenda. There’s a presentation of where we are in terms of the objectives and mission of the NAP. Where we are, then we try to ask from the sectors the constraints and problems they have been encountering regarding AMR.”*
(Animal Health—07, National Organization)

Outside of the national government, a few respondents reported that, although data collection is conducted at the hospital level and supervised by the sub-national health office, there was no feedback provided to sub-national offices apart from the publication of the results in an annual report.

### 3.4. Sustainability

Sustainability was an issue viewed with both concern and optimism. Some stakeholders argued that sustainability was possible with more awareness, better regulatory oversight, as well as a level of behavior change among the population, but others viewed it as a major setback, with AMR being less prominent in a health system struggling to address the COVID-19 pandemic. It was suggested that a review of the global strategy presents an opportunity to recalibrate and build on the initial successes and momentum of the national plan and program. A respondent cited that good health outcomes of constituents were a major driver for political will.


*“So, nobody will not follow because all these local officials, you know what’s important for them—they want to win the elections. So if you won’t take good care of the health of your people, then you will not win.”*
(Human Health—12, International NGO)

Similar to other program activities, funding for AMR remained a significant challenge. Despite their limited funding streams, ICAMR agencies strived to allocate budgets for their AMR activities. Several respondents observed that the roles and impacts of the animal and environmental health sectors on the issue of AMR were not the main focus of decision-makers, contributing to the continued neglect and underdevelopment of this component of the AMR program.

### 3.5. One Health Engagement

With a few exceptions, the stakeholders were aware of the One Health approach and its importance in addressing AMR. They mentioned that it was fundamentally about collaboration and involvement within and outside the three sectors. Several respondents observed that this concept seems to resonate strongly with practitioners, some of whom belong to the Philippine One Health Network. Others pointed out that the human and animal sectors appear to be doing better at integration, even if One Health was not as prominent in the discourse or practice of healthcare professionals. The environment sector, however, was missing from the equation. This seeming detachment was a common observation from the respondents in other sectors working on AMR.


*“For the animals and humans, the integration is good. I see some positive signs in terms of integrating humans and animals, but the environment side is detached, it is not yet fully integrated.”*
(Animal Health—07, National Organization)

## 4. Discussion

This study has detailed the development and implementation of the NAP on AMR in the Philippines following an AMR governance framework. Our respondents highlighted areas that have been executed well, including (1) a high level of governmental support and involvement of relevant stakeholders, (2) the development of policies to support improved responses in infection prevention and control (IPC) and antimicrobial stewardship (AMS), and (3) better engagement and advocacy by professional associations and civil society groups. However, there were some areas that were lacking and could be further improved. These included (1) a lack of resources and regulatory capacity, (2) insufficient impetus for AMR research and surveillance, and (3) limited One Health engagement.

A high level of governmental support and the involvement of relevant stakeholders provided an enabling environment for AMR policies. Technical assistance and funding from the tripartite, particularly in the initial phases of research and policy development, strengthened the evidence base and capacity for improved AMR responses at the national level [16]. A country situation analysis on the AMR situation in the Philippines was conducted in 2013 when the WHO and other stakeholder groups considered related research on health facilities and the pharmaceutical sector, respectively [14]. This assessment, based on the six-point policy package of the WHO, provided details that included the prevalence of AMR organisms in the country, which, in turn, informed stakeholders regarding the appropriate objectives to be included in the NAP [22]. Subsequently, policy reforms were introduced through the Inter-agency Committee on AMR (ICAMR), which led to better One Health engagement to help achieve targets under the NAP. Our study respondents also highlighted a high level of participation throughout the development of the NAP. The first NAP took at least two years to develop as it involved stakeholders from the government (led by the Department of Health (DOH) and the Department of Agriculture (DA)), professional societies, industry associations, hospitals, and academia. An implementation review conducted in parallel with the development of the 2019 NAP was also guided by similar stakeholders. The analysis of NAPs by Chua et al. revealed varying levels of stakeholder participation across the ASEAN member states, with countries such as Singapore acknowledging a need for better engagement from a wide range of stakeholders, including those from the community [17]. This issue was also highlighted in other similar studies [23,24].

The study respondents have noted that, over the years, there were improved responses in the areas of IPC and AMS in human health, mainly because of enabling policies that institutionalized these activities in the hospitals. Part of the response to improve IPC involves the establishment of the Infection Control Committee, which was composed of management staff, physicians, and other healthcare workers from clinical microbiology, pharmacy, sterilizing service, housekeeping, and training services. This multidisciplinary committee ensured the implementation of infection control strategies by formulating and updating infection control policies, guidelines, and procedures. The National Standards in Infection Control for Healthcare Facilities were published in 2009 to strengthen the IPC programs nationwide and prevent healthcare-associated infections, highlighting criteria, including IPC training and related indicators, as part of the hospital licensing requirements. In animal health, the harmonization of IPC standards and practices was being pursued at the ASEAN level. The AMS activities started in the Philippines with support from the WHO. This was subsequently expanded to the national level with the launch of the national AMS program in 2015 [15]. In addition, the DOH was tracking whether hospitals had fully implemented AMS programs [20]. These strategies were only implemented in the human health sector and should be replicated in the animal health and environmental sectors through policy and investment support. At the global level, AMS in human health was more established than in animal health, which was gaining prominence in recent years, and the environment sector [25,26].

Better engagement and advocacy by professional associations and civil society groups have been highlighted as factors that are relevant to increased efforts regarding education and public awareness, as well as the optimization of antimicrobial usage. Our respondents reported that various medical practitioners, academic institutions, and patient groups are doing their part in curbing AMR either by their own initiatives or participating in ICAMR-led awareness efforts. For instance, AMR has been included in continuing professional development programs among professional societies, and pharmacists and drug stores have been ensuring the implementation of the FDA policy of “no prescription, no dispensing” for antimicrobials and the Philippine Pharmacy Act (Republic Act No. 10918). The increased level of engagement was also observed in the animal health sector, as is evident in the “IAMResponsible” campaign for the animal and agricultural sector, specifically targeting veterinarians, farmers, students, and the general public. This finding is similar across other ASEAN countries and could likely be attributed to the increased awareness and advocacy at the global level through international initiations, such as the WHO’s World Antimicrobial Awareness Week in November annually [17,27]. However, some respondents noted that the level of attention on the issue tends to wane after these campaign activities. They also highlighted that having indicators of effectiveness could help in the planning and development of messages and how best to engage target stakeholders.

The respondents highlighted that there was a lack of resources and capacity, which could have impeded the progress of implementation plans. Although more emphasis has been placed on AMR recently, it was still reported to be low on the hierarchy when benchmarked against other issues at the national level. For example, at the DA, funding was mostly for crops, such as rice and corn, and animals accounted for a small proportion of the funds. In the Inter-agency Committee on Environmental Health, in which the DOH and the Department of Environment and Natural Resources were represented, there is no mention of AMR in the main policy areas. A study suggested that, although external funding support from international organizations helped initiate some of the plans, sustainability was largely dependent on internal government funding and sustained support from policymakers [28]. Besides funding, there was also a constraint on human resources. The wide scope of AMR made it especially difficult to allocate resources, including specialists in the field of infectious diseases, as well as clinical pharmacists specifically for AMS. Agencies such as the FDA were similarly confronted with the lack of qualified human resources to monitor regulations to combat AMR.

The research and surveillance on AMR could be improved in the Philippines. Our respondents highlighted that, overall, there were very few specialists and researchers focusing on AMR. In addition, although there was research funding available from agencies, such as the Department of Science and Technology, the funding allocation for research was largely based on approved research grants that were subject to competition from other non-AMR topics. Other studies have also reported a lack of adequate capacity to support research and development in similar settings [29,30]. Besides research, surveillance was also reported to be insufficient. The respondents observed that a limited capacity of trained personnel and available diagnostics resulted in a small number of sentinel sites; in particular, the sentinel-based surveillance does not include private health facilities. This was also echoed in the animal health and environment sectors. Potential laboratory facilities for both animals and the environment were already overwhelmed with surveillance for AMR in the human sector. Insufficient infrastructure to support a national surveillance system has also been reported by other countries [29]. For example, Thailand highlighted a lack of national policy decisions for the development of a comprehensive surveillance system; Singapore mentioned that, although surveillance in the human health sector was well established, it was mainly limited to the public sector, and it was also lacking in the animal health and environment sectors [27,31]. The antimicrobial usage (AMU) surveillance in the Philippines was also reported to be insufficient. The AMU data in the Philippines were based on sales and distribution data, which were not equivalent to the more accurate consumption data. There have been suggestions to expand surveillance with the inclusion of additional private hospitals and strengthening the capacity for animal and environmental surveillance to reflect a more accurate picture of the AMR situation nationally. Overall, there should be a tie-in between the surveillance systems and research to inform the emergence and evolution of AMR and to support the development of intervention strategies through collaboration with international and regional bodies where possible [29,30].

Finally, although One Health engagement has been mentioned, the respondents felt that it was limited. Apart from the work of the ICAMR, the One Health approach was not fully realized in the agency-led programs and coordination mechanisms. The interventions and resources remain sector-specific rather than integrated or multisectoral in implementation. This could be due to a varied representation and involvement in AMR across sectors. Specifically, the human health sector was relatively more established as compared to the animal health and environmental sectors. The respondents highlighted two possible reasons for this: (1) incomplete and insufficiently well-defined roles of different stakeholders in the NAP, and (2) differences in priorities for AMR among the various sectors. Cross-sectoral coordination has been highlighted to be difficult in other countries due to fragmented and inflexible structures for bridging sectors [24,32,33]. In addition, a respondent highlighted that implementation follows a top-down approach rather than involving more local/sub-national stakeholders. For example, the reporting of AMR surveillance data has been described as a one-way process, with hardly any feedback returning from the national level down to regional and local implementers. Overall, there should be the involvement of more stakeholders in the AMR program. There should be the inclusion of environmental representatives in AMR policy efforts to help establish AMR as a priority in environmental policies and programs; involvement of the local/sub-national stakeholders, such as those in the local government units, in implementation and regulation activities; and harmonization of policies and priorities to support the sustainability of AMR initiatives.

Based on our study results, recommendations from our respondents, and findings from international literature, we came up with a package of policy recommendations in Table 3 [30,32,33,34,35].

To our knowledge, this is one of the first studies to explore the development and implementation of the AMR NAP in the Philippines. We utilized Chua’s AMR governance framework to guide our analysis, which allowed us to review our data in a structured manner. We were able to recruit and gather the perspectives of multiple stakeholders from the human health, animal health, and environmental sectors. There were some limitations. Firstly, the online interviews conducted due to COVID-19 restrictions could have constrained personal interactions, often crucial in maximizing the value of interviews. For sampling, even with due diligence to ensure that the respondent is the best authority to represent a perspective or opinion, the extent of their actual involvement in the policy process of developing and implementing the NAP on AMR is beyond our control. The representation from the animal health and environment sectors was much less compared to that from the human health sector. However, this was expected given that the DOH was much more involved and invested compared to the other agencies.

## 5. Conclusions

The magnitude of the AMR problem has prompted a high-level response from the national government and led to the creation of the ICAMR. Although there has been considerable progress for human health, strengthening the involvement and representation of the animal health and environment sectors in the AMR scene must be undertaken. The development of well-defined roles within policies will be paramount to the strong implementation of strategies to combat AMR. The enhancement of IPC strategies, surveillance and research capacities, AMS, and education and awareness campaigns across all three sectors will contribute to establishing a robust and comprehensive AMR program in the Philippines.

## Figures and Tables

**Figure 1 antibiotics-11-00820-f001:**
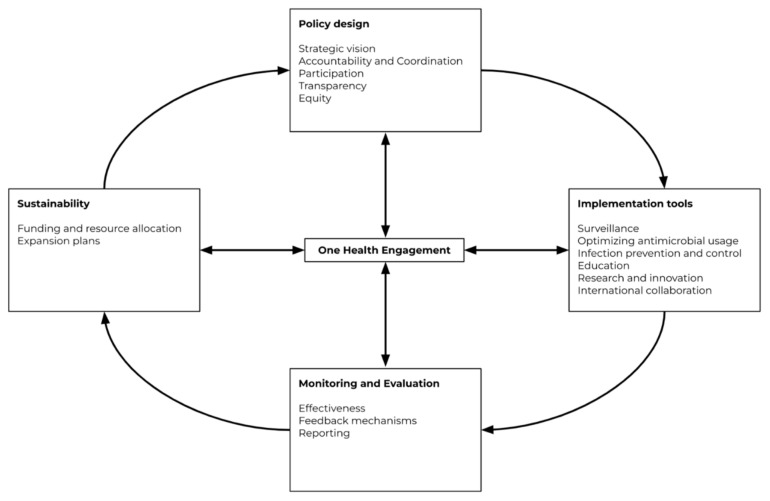
Governance framework for assessment of national action plans on antimicrobial resistance from Chua et al., 2021 [17]. Reuse is licensed under an open access Creative Commons CC BY 4.0 license.

**Table 1 antibiotics-11-00820-t001:** Summary of respondents by type of institution and sector.

Type of Institution	Sector				Total
	Human Health	Animal Health	Environment	Human and Animal Health	
Academia	2	3	4	-	9
Government (National)	4	-	-	1	5
Government (Sub-national)	6	-	-	-	6
International Non-governmental Organization	4	1	1	-	6
National Organization	9	2	-	-	11
Total	25	6	5	1	37

**Table 2 antibiotics-11-00820-t002:** Philippine National Action Plan targets for 2019–2023 [16].

Target	Details
1	Reduce by 10% carbapenem-resistant Enterobacteriaceae (*Escherichia coli* and *Klebsiella* spp.) infections acquired during hospitalization
2	Maintain the 0% prevalence of ceftriaxone-resistant *Neisseria gonorrhoeae*
3	Reduce by at least 10% the overall methicillin resistance in *Staphylococcus aureus* bloodstream infections compared to rates in 2017
4	Reduce by 10% multidrug-resistant *Pseudomonas* spp. infections acquired during hospitalization compared to estimates in 2017
5	Reduce by 25% ciprofloxacin-resistant non-typhoidal Salmonella infections compared to 2017
6	Reduce by 10% use of antibiotics in humans and animals
7	Identify baseline AMR and use in animal sector

AMR = antimicrobial resistance.

**Table 3 antibiotics-11-00820-t003:** Policy recommendations.

S/N	Policy Recommendations	Details
1	Active engagement of other ICAMR agencies	More active participation from DTI, DILG, and DENR will provide a more holistic approach to tackle AMR.
2	More involvement of stakeholders beyond the policymaking level	LGUs should be empowered to implement AMR strategies at the local level. Stakeholders on the ground, such as community pharmacists and those from the backyard farm sector, should be represented during policy discussions.
3	Define clear roles for all relevant stakeholders	Well-defined roles will help improve accountability and coordination required for effective implementation of all aspects of the AMR program. Specifically, greater emphasis must be made in defining the roles of the local governmental units and civil societies.
4	Improve research capacity	AMR research serves to bridge knowledge gaps and provides evidence to build governmental support for development of better policies on AMR.
5	Develop better surveillance systems	The current AMR surveillance system is limited and should be extended to more surveillance sites, including private healthcare facilities, as well as areas in the animal and environmental sectors. This should be supported by strengthening of laboratory infrastructures and capacity. AMU surveillance systems should be established to better assess the level of AMU nationally.
6	Improve coordination across sectors through a One Health approach	Better integration across sectors is crucial to facilitate implementation of the NAP. For example, an integrated AMR surveillance system will provide a clearer idea of the actual AMR situation in the country, possibly elucidating the transmission pathways of certain microorganisms.
7	Strengthen partnerships with organizations at the regional and international level	International collaboration supports knowledge generation and innovation, as well as provides an avenue for better resources and capacity to execute the implementation plans.
8	Develop monitoring and evaluation systems for all implementation plans	Monitoring and evaluation are key in determining effectiveness of a plan and providing evidence to inform policies. Regular monitoring and evaluation activities will advise if the implementation plans are on track or if adjustments must be made.
9	Set intermediate targets for the implementation plans	These intermediate targets will pave a structured and systematic path towards programmatic success.

ICAMR = inter-agency committee on antimicrobial resistance; DTI = department of trade and industry; DILG = department of the interior and local government; DENR = department of environment and natural resources; AMR = antimicrobial resistance; LGUs = local government units; AMU = antimicrobial use; NAP = national action plan.

## Data Availability

Not applicable.

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
