# Peer review of "A Qualitative Study on the Design and Implementation of the National Action Plan on Antimicrobial Resistance in the Philippines"

_antibiotics, 2022, doi:10.3390/antibiotics11060820_

Round 1
Reviewer 1 Report
The study is interesting but does not report the research methodology or results in quantitative terms. The statistical methodology and quantitative description of the results are completely missing. The study cannot be accepted in this version.
Author Response
General Comment:
The study is interesting but does not report the research methodology or results in quantitative terms. The statistical methodology and quantitative description of the results are completely missing. The study cannot be accepted in this version.
Response:
This study utilized a qualitative design at the onset. The qualitative methodology was similar to previously published AMR policy studies (https://doi.org/10.3390/antibiotics10080991, https://doi.org/10.3390/antibiotics8040201). Based on the study objectives, there is no need to use quantitative design.
The data analysis adopted an interpretative approach which focuses on participants’ perceptions and interpretations of the discussion topic. We used Chua’s AMR governance framework to deductively guide our coding process, while allowing codes that did not fit into the framework to develop as emerging themes inductively. In addition, our study was reported according to the COREQ checklist, which was intended to promote explicit and comprehensive reporting of qualitative studies.
The revised manuscript is also attached for your reference.
We hope this is clear and satisfactory for the reviewer. We are, of course, happy to make any further changes as suggested subsequently.
Please see attachment.

Reviewer 2 Report
The present study is a qualitative analysis of the AMR national action plan in the Philippines among 37 participants from the One Health spectrum. Thematic synthesis was performed on enabling factors and challenges, and overall, the study found a variety of interesting facilitators and barriers that may allow for future actions towards strong implementation of AMR strategies. These data are informative and interesting. I applaud the efforts of the authors in their work and recommend acceptance for publication
Author Response
General Comment:
The present study is a qualitative analysis of the AMR national action plan in the Philippines among 37 participants from the One Health spectrum. Thematic synthesis was performed on enabling factors and challenges, and overall, the study found a variety of interesting facilitators and barriers that may allow for future actions towards strong implementation of AM strategies. These data are informative and interesting. I applaud the efforts of the authors in their work and recommend acceptance for publication.
Response:
Thank you for the compliment and for recommending acceptance of our manuscript. We appreciate your time taken to review our manuscript.
Reviewer 3 Report
The authors have presented a study aimed at qualitatively evaluating the design and implementation of the Philippine Antimicrobial Resistance National Action Plan. Given the threat that antimicrobial resistance poses to global public health and the environment, studies of this type are needed. The effort by the authors would add valuable information to the field of antimicrobial resistance.
Abstract, Title, References:
The aim of the study is clear. It is also clear what the study found and how the authors did it. The title communicates the intent of the authors clearly and succinctly and is informative and relevant. The references are relevant, recent, referenced correctly, and appropriate key studies are included.
Introduction:
The authors have clearly described what was previously known about the topic and the research is clearly outlined. Given what is already known and the gaps in our knowledge, the research question is justified.
Materials and Methods:
This section should be clearer. Of the 37 respondents, who responded to what? Uncertain what the variables are and how they were measured/defined. Uncertain if there is enough detail to replicate the study.
Results:
The authors presented the data in a manner that invites many questions even if the study is a qualitative one. For example,
- The quotes from the respondents do not tell how many of them responded a certain way.
- Instead of using quotes, could authors give a percentage of respondents who gave answers that were similar/different? Maybe a table describing the responses?
- How did the authors account for biases among respondents? This seems to be lacking.
Discussion and Conclusions
The Results are discussed in the context of the experimental design. However, as discussed in the comments for said Results, the quotes from unknown number of respondents and not accounting for respondent biases are problematic.
There are several concerns that the authors must address as follows:
- What are the limitations of the study?
- How reproducible is the study?
- The small sample size (n=37) makes it difficult to draw generalizable conclusions because the data may be biased and unrepresentative of the wider population. Please address.
- The authors used a lot of quotes but did not indicate how many respondents gave the same answers, thus making the claims somewhat unreliable.
Other points needing clarification:
Line 114: How appropriate is the sample size? Did authors determine whether the sample size was appropriate to draw generalized conclusions? At first glance, the number of participants (n=37) appears to be insufficient to be statistically significant. However, per the Central Limit Theorem (CLT), sample sizes equal to or greater than 30 are often considered sufficient for CLT to hold. Did the authors consider that during experimental design?
Lines 204-205: Which FDA are authors referring to? The US FDA? Please clarify.
Line 315: Please add "of" to the sentence to read, "The components "of" the plan..."
Line 442: Please spell out ASEAN – assuming it’s Association of Southeast Asian Nations
Line 444: Please clarify the phrase "has been cited." Who cited and where?
Lines 776-778: This sentence is incomplete. Suggest combining with previous sentence.
Round 2
Reviewer 1 Report
The clarifications provided by the authors satisfy previous requests. Indeed, it has been clarified that this is a qualitative and not a quantitative research. The statistical methodology used may be sufficient to answer the study question. The topic is interesting and I propose publication of the manuscript
Author Response
Thank you for your final approval of our study.
Reviewer 3 Report
Dear Authors,
Thank you for taking the time to address my concerns. You have addressed them adequately and overall, the manuscript has been much improved.
Author Response

(The authors gave the same response as above.)
